# Endobronchial ultrasound-guided transbronchial needle aspiration versus mediastinoscopy for mediastinal staging of lung cancer: A systematic review of economic evaluation studies

João Pedro Steinhauser Motta[1]*, Ricardo E. Steffen[2], Caroliny Samary Lobato[3], Vanessa Souza Mendonça[4], José Roberto Lapa e Silva[1]

**1** Instituto de Doenças do Torax, Universidade Federal do Rio de Janeiro, Rio de Janeiro, Brazil, **2** Instituto de Medicina Social, Universidade Estadual do Rio de Janeiro, Rio de Janeiro, Brazil, **3** Programa de Pós-Graduação em Clínica Médica da Faculdade de Medicina, Universidade Federal do Rio de Janeiro, Rio de Janeiro, Brazil, **4** Biblioteca do Hospital Universitário Clementino Fraga Filho, Universidade Federal do Rio de Janeiro, Rio de Janeiro, Brazil

☯ These authors contributed equally to this work.
‡ These authors also contributed equally to this work.
* joaosteinhauser@gmail.com

## Abstract

### Introduction

The emergence of endobronchial ultrasound (EBUS) changed the approach to staging lung cancer. As a new method being incorporated, the use of EBUS may lead to a shift in clinical and costs outcomes.

### Objective

The aim of this systematic review is to gather information to better understand the economic impact of implementing EBUS.

### Methods

This review is reported according to the PRISMA statement and registered on PROSPERO (**CRD42019107901**). Search keywords were elaborated considering descriptors of terms related to the disease (lung cancer / mediastinal staging of lung cancer) and the technologies of interest (EBUS and mediastinoscopy) combined with a specific economic filter. The literature search was performed in MEDLINE, EMBASE, LILACS, Cochrane Library of Trials, Web of Science, Scopus and National Health System Economic Evaluation Database (NHS EED) of the Center for Reviews and Dissemination (CRD). Screening, selection of articles, data extraction and quality assessment were carried out by two reviewers.

**Data Availability Statement:** All relevant data are within the manuscript and its Supporting Information files.

**Funding:** Funding by this study was provided by the Higher Education Personnel Improvement Coordination (CAPES) of the Brazilian Ministry of Education (CAPES AUXPE PROEX 0641/2018). The funders had no role in study design, data collection and analysis, decision to publish or preparation of the manuscript.

**Competing interests:** The authors have declared that no competing interests exist.

## Results

Seven hundred and seventy publications were identified through the database searches. Eight articles were included in this review. All publications are full economic evaluation studies, one cost-effectiveness, three cost-utility, and four cost-minimization analyses. The costs of strategies using EBUS-TBNA were lower than the ones using mediastinoscopy in all studies analyzed. Two of the best quality scored studies demonstrate that the mediastinoscopy strategy is dominated by the EBUS-TBNA strategy.

## Conclusion

Information gathered in the eight studies of this systematic review suggest that EBUS is cost-effective compared to mediastinoscopy for mediastinal staging of lung cancer.

## Introduction

Lung cancer is a major health problem, with estimates of 155.870 deaths in the United States in 2017 [1] and 1.6 million tumor-related deaths annually worldwide [2]. Except for a proportion of patients diagnosed at the early stage of the disease or others with known distant metastasis, many of the patients with lung cancer will have an indication of an invasive staging of the mediastinum [3–5]. The emergence of endobronchial ultrasound (EBUS) [6], a minimally invasive procedure capable of providing valuable information for primary tumor diagnosis and mediastinal staging [7–9], significantly changed the approach to staging lung cancer, becoming part of the routine mediastinal evaluation of lung cancer in developed countries [10, 11]. A recent systematic review and meta-analysis of randomized controlled trials and observational studies comparing EBUS with mediastinoscopy suggested an equivalence of the two procedures for mediastinal staging of lung cancer, with a lower complication rate favoring the endosonographic approach [12].

In an era of increasing cost pressures, restructuring of health care delivery and payment, and heightened consumer demand, technology can be managed in ways that improve patient access and health outcomes, while continuing to encourage useful innovation [13]. As a new method being incorporated by different health systems, the use of EBUS may lead to a shift in clinical and costs outcomes. An important question to be answered at this point is: is the use of EBUS for the mediastinal staging of lung cancer cost-effective when compared to mediastinoscopy? Some economic evaluation studies published in the last 10 years have analyzed the incorporation of the EBUS technique in different health systems [14], but until now the cost-effectiveness of EBUS versus mediastinoscopy has not been demonstrated in prior clinical trials. The primary objective of this study is to understand the cost-effectiveness ratio of EBUS compared to mediastinoscopy for invasive mediastinal staging of lung cancer. Secondary objectives are to identify the most relevant studies published on the topic and the types of models used in those publications, to understand the most important economic trade-offs and to guide future economic assessments on this topic in countries with different health systems.

## Material and methods

This systematic review is reported according to the PRISMA statement [15]. A protocol of the review was registered on PROSPERO (International Prospective Register of Systematic Reviews), registry number **CRD42019107901** and published previously [16]. An ethics

committee approval was not required as this is a systematic review of published data, with no exposure of individual patient data.

## Research problem, search keywords and bibliographic search

The PICO strategy was used to formulate the research problem. The search keywords were elaborated considering descriptors of terms related to the disease (lung cancer / mediastinal staging of lung cancer) and the technologies of interest (EBUS-TBNA and mediastinoscopy) combined with a specific economic filter (search strategy of the Canadian Agency for Drugs and Technologies in Health—CADTH) [17]. The literature search was divided into 3 parts: 1) Search the PROSPERO platform for systematic reviews on this subject already published or in progress 2) Search in electronic databases: MEDLINE (Pubmed), EMBASE, LILACS, Cochrane Library of Trials, Web of Science, Scopus, National Health System Economic Evaluation Database (NHS EED) of the Center for Reviews and Dissemination (CRD) 3) Cross-analysis of the bibliographic references of the articles selected in the database search phase. The authors chose not to include unpublished data and gray literature in the searches. The PRISMA checklist, search keywords used and the search strategy used for Medline (Pubmed) can be accessed in the supporting information session of this article. Studies obtained from the search strategy were sent to a reference management tool (EndNote X8®—Clarivate Analytics—Philadelphia—USA) to identify and eliminate duplicate references.

## Screening and selection of articles, data extraction and quality assessment

Screening, selection of articles, data extraction and quality assessment were carried out by two independent reviewers (JPSM and CSL–screening and selection of articles / JPSM and RES– data extraction and quality assessment). Discrepancies between the two reviewers were resolved by consensus. Inclusion criteria were: Articles in English, German, Spanish and Portuguese language; full economic evaluation studies; studies on the mediastinal staging of lung cancer. Exclusion criteria were: studies not focused on EBUS and mediastinal lung cancer staging; annals of congress, editorials, letters or review articles; partial economic evaluations. A structured data abstraction form was used and can also be accessed in the supporting information section of this article. For each included paper, data relating to the identification, type of economic evaluation, study design, population, study perspective, time horizon, intervention and comparators, measures of effectiveness, measures of costs, discount rate, model used, outcomes, sensitivity analysis, cost-effectiveness threshold, conclusions and other relevant characteristics were extracted. The quality assessment tool used was the Consolidated Health Economic Evaluation Reporting Standards (CHEERS) [18]. Costs of EBUS-TBNA and mediastinoscopy were updated to 2018 values and converted to the international dollar (I$) to help comparison between different studies. We used the method suggested by Turner on recent publication [19], adjusting local inflation rates by the Gross Domestic Price (GDP) implicit price deflator, then converting to I$ with the exchange rate of 2018. Currency conversion data were extracted from the World Bank webpage (data.worldbank.org). None of the authors was contacted for further clarifications.

## Results

Fig 1 illustrates the flow of identification and selection of articles. No published or ongoing systematic review on this topic was found on the PROSPERO platform. Seven hundred and seventy publications were identified through the electronic database searches. Two hundred and forty-three duplicate reports were excluded and 509 were removed after title and abstract screening. Twenty publications were assessed for eligibility based on full-text, of which 8

articles were finally included in this review. Cross-analysis of references identified a cost-effectiveness study on the subject of this review published by Rintoul in 2013 [20]. However, because it is a study published by the same group and addressing the same clinical trial

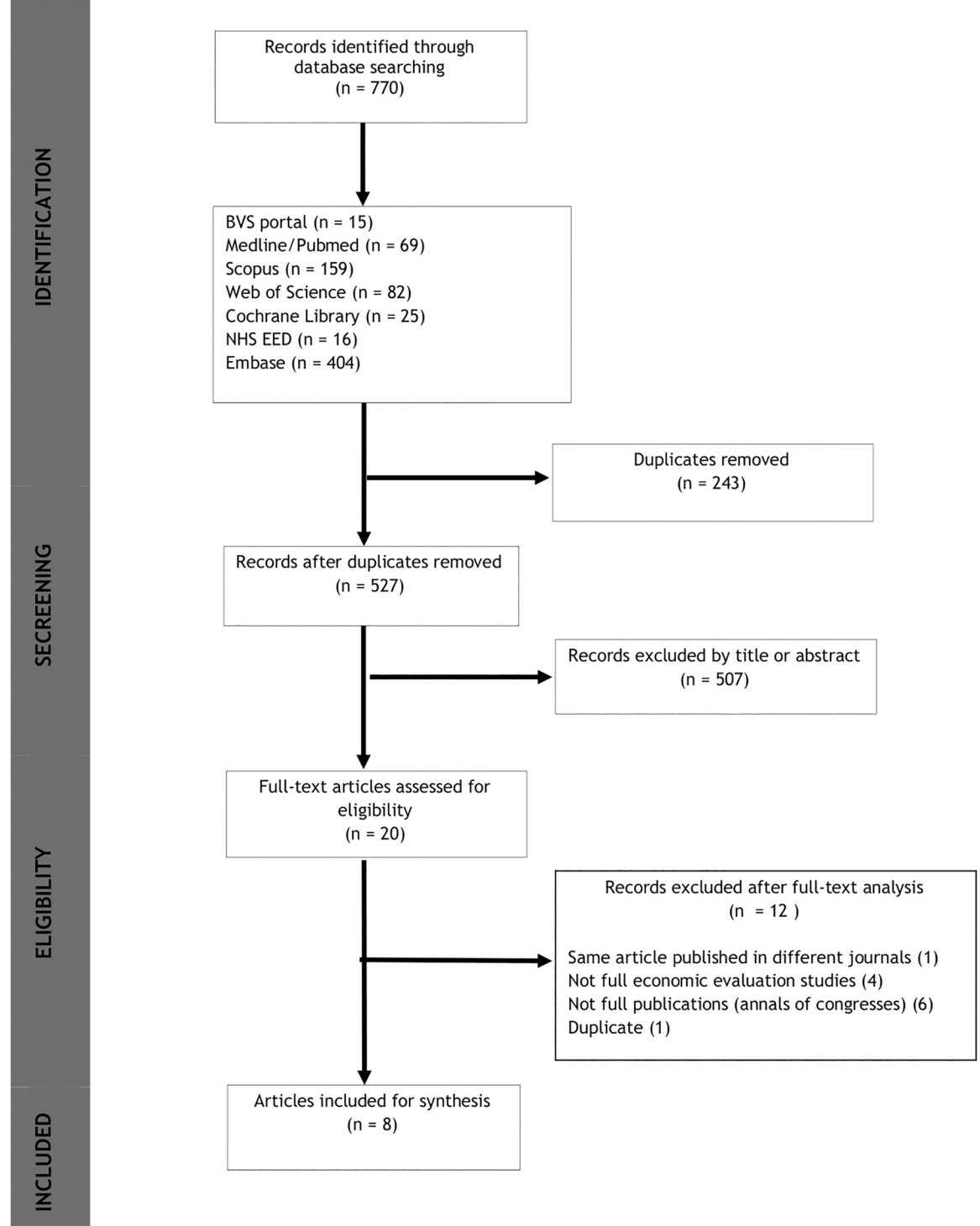

**Fig 1. Flow of identification and selection of articles.** BVS, Biblioteca Virtual de Saúde; NHS EED, National Health System Economic Evaluation Database.

**Table 1. Study characteristics.**

| Author | Year | Country | Type of Evaluation | Population / N | Type of Study | Model | Perspective | Time Horizon | Sensitivity Analysis |
|--------|------|---------|--------------------|----------------|---------------|-------|-------------|--------------|----------------------|
| Ang et al [28] | 2010 | Singapore | Cost-minimization | NSCLC requiring mediastinal staging / N.A. | M | Decision tree | Hospital | N.A. | One-way and two-way |
| Czarnecka-Kujawa et al [26] | 2016 | Canada | Cost-utility | Verified or suspected NSCLC clinical N0 / N.A. | M | Decision tree | Health Care System | lifetime | One-way and two-way |
| Harewood et al [25] | 2009 | USA | Cost-minimization | Verified or suspected NSCLC after chest CT / N.A. | M | Decision tree | Payers Perspective | N.A. | One-way and two-way |
| Luque et al [22] | 2016 | Spain | Cost-utility | NSCLC without distant metastases / N.A. | M | Influence Diagram | Health Care System | N.A. | Multi-way |
| Navani et al [23] | 2012 | United Kingdom | Cost-minimization | Isolated mediastinal lymphadenopathy on CT or PET-CT / 77 | T/M | Decision tree | Health Care System | N.A. | N.A. |
| Sharples et al [21] | 2012 | Belgium, Netherlands, and the UK | Cost-utility | Operable NSCLC requiring mediastinal staging / 241 | T | N.A. | Health Care System | 6 months | One-way |
| Søgaard et al [22] | 2013 | Denmark | Cost-effectiveness | Verified operable NSCLC / N.A. | M | Decision tree | Health Care System | 5 years | One-way |
| Steinfort et al [27] | 2010 | Australia | Cost-minimization | NSCLC requiring mediastinal staging after PET-CT / N.A. | M | Decision tree | Hospital | N.A. | One-way and two-way |

USA, United States of America; NSCLC, non-small cell lung cancer; CT, computed tomography; PET, positron emission tomography; M, model-based; T, trial-based; N, number of patients; N.A., not available

(ASTER trial) as the Health Technology Assessment (HTA) published by Sharples already included in this review [21], this article was excluded.

## Study characteristics

Table 1 summarizes the characteristics of the eight studies included in this systematic review. The articles were published between 2009 and 2019. Five studies were based on European countries [21–24], one from the United States [25], one from Canada [26], one from Australia [27] and one from Singapore [28].

The eight publications are full economic evaluation studies, one of them is a cost-effectiveness study [24, 28], three are cost-utility analyses [21, 22, 26] and four are cost-minimization analyses [23, 25, 27, 28]. Six studies are model-based, five of them used a decision tree analysis model [24–28] and one used an influence diagram [22]. Prevalence of N2/N3 disease and sensitivity of EBUS-TBNA and mediastinoscopy were the main parameters used on the models. The data sources for these parameters were based mostly on systematic reviews, but also from clinical trials [25], observational studies [27] and national registries [24]. Two studies are trial-based [21, 23]. The HTA report published by Sharples et al is an economic analysis of the Assessment of Surgical Staging versus Endoscopic Ultrasound in Lung Cancer (ASTER) trial, a randomized controlled trial that compared endosonographic against surgical staging for patients with potentially operable lung cancer [21]. Navani et al published a prospective multi-center trial with a cost-analysis of 77 consecutive patients with isolated mediastinal lymphadenopathy that underwent EBUS-TBNA [23]. The populations considered in the studies did not differ considerably. Six articles [21, 22, 24, 25, 27, 28] evaluated patients with suspected or diagnosed lung cancer and indication for invasive mediastinal staging. Czarnecka-Kujawa et al [26] focused specifically on patients with clinical N0/N1 status based on chest computed tomography (CT) and positron emission tomography-computed tomography (PET-CT)

and Navani [23] et al discussed the use of EBUS in patients with isolated mediastinal lymph-adenopathy (not necessarily lung cancer). A health system perspective was adopted in 75% of the publications [21–24, 26, 27]. Harewood et al presented a payers' perspective based on Medicare reimbursement rates [25], Ang et al and Steinfort et al adopted a hospital perspective [28]. The time horizon of the economic evaluation was reported in three studies, Sharples et al [21], Søogard et al [24] and Czarnecka-Kujawa et al [26]; considering six months, five years and lifetime horizon respectively. EBUS-TBNA was the major intervention and the compara-tor was mediastinoscopy in four publications [21, 23, 26, 28]. The other studies included EBUS as one of several possible mediastinal staging strategies [22, 24, 25, 27]. The comparators varied from just mediastinoscopy or mediastinoscopy and other staging modalities, such as blind transbronchial needle aspiration (blind-TBNA), endoscopic ultrasound (EUS), PET-CT and chest CT. The sensitivity of EBUS-TBNA was the most common parameter for effectiveness. A sensitivity analysis of the results was performed by all model-based studies but is not in one of the trial-based studies [23]. The models were tested mainly concerning variations in the sensi-tivity of EBUS-TBNA and the prevalence of N2 / N3 disease in the study population.

## Cost data

Table 2 summarizes the cost data. Only direct medical costs were presented, none of the stud-ies reported indirect costs. In five from the eight publications disaggregated cost items [22, 23, 25, 27, 28] were not reported and it is not clear which items were included. Most costs refer only to procedures and their complications. Noteworthy is the HTA published by the ASTER Trial Group, detailing costs related to staff time, bed occupancy rates, hospital fees, equipment costs (five-year lifetime), consumables, sterilization of scopes and maintenance contracts [21]. Cost data sources varied from local hospital primary data [26–28], secondary data from pub-lished literature [22], to national tariffs and/or Diagnosis-Related Groups (DRG) fees [21, 23–25]. To allow the comparison of costs related to procedures or strategies involving the use of EBUS-TBNA and mediastinoscopy, we used the method proposed by Turner [19], with infla-tionary adjustment and conversion of costs to I\$ for all publications. Despite the different cost items included by the authors, in all cases the costs of strategies using EBUS-TBNA were lower than the ones using mediastinoscopy.

## Interventions, comparators and outcomes

Table 3 summarizes the interventions, comparators and outcomes used in each study. Four publications (cost minimization studies) [23, 25, 27, 28] estimated cost-savings associated with the use of EBUS-TBNA compared to mediastinoscopy and other staging techniques. Sensitiv-ity analyses were carried out in three from the four cost minimization publications [25, 27, 28]. Considering the other 4 studies (cost-utilities and cost-effectiveness analyses), Czarnecka-Kujawa et al reported the incremental cost-effectiveness ratio (ICER) for different approaches for clinical N0/N1 lung cancer [26], Sharples et al searched for cost-utility regarding a strategy of endosonographic followed by surgical staging (in case of negative findings) compared to surgical mediastinal staging for patients with potentially operable lung cancer [21], Søogard et al calculated costs for life-years gained comparing six distinct strategies for patients with his-tologically proven NSCLC [24], and Luque et al reported the result of their study as an optimal sequence of tests for mediastinal staging [22].

## Cost-effectiveness, cost-savings and sensitivity analysis results

Table 4 shows the results of cost-effectiveness, cost-savings, sensitivity analysis and conclu-sions as described in each study. The 4 cost-minimization studies [23, 25, 27, 28] demonstrated

**Table 2. Cost data.**

| Author | Type of Costs | Cost Items | Cost Data Sources | Year Accounted | Currency Unit | Inflation Rate | Discount Rate | *Willingness-to-pay Threshold | *EBUS-TBNA $ | *Mediastinoscopy $ |
|---|---|---|---|---|---|---|---|---|---|---|
| **Ang et al [28]** | Direct medical costs | Facility fees, manpower and consumables | Primary data (average full-fee paying bills from Singaporean General Hospital) | 2009 | Singaporean Dollar (SGD) | N.A. | N.A. | N.A. | SGD$ 2.623 Int$ 2.478 | SGD$ 3.007 Int$ 2.841 |
| **Czarnecka-Kujawa et al [26]** | Direct medical costs | Average procedures costs, costs of complications, cost of chemotherapy and radiotherapy | Primary data (recorded hospital costs from the Toronto General Hospital between 2005–2014) | 2015 | Canadian Dollar (CAD) | adjusted to 2015 | N.A. | CAD$ 80.000/ QALY Int$ 99.920/QALY | CAD$ 13.727 Int$ 18.026 | CAD$ 18.143 Int$ 23.816 |
| **Harewood et al [25]** | Direct medical costs | Facility and professional fees (outpatient) DRG for NSCLC (inpatient) | Medicare ambulatory patient classification (outpatient) Medicare pays based on DRG for patient with NSCLC (inpatient) | 2007 | US Dollar (USD) | N.A. | N.A. | N.A. | USD$ 19.828 Int$ 23.595 | USD$ 20.157 Int$ 23.986 |
| **Luque et al [22]** | Direct medical costs | Procedures costs | Secondary data from published literature | 2010 | Euro (EUR) | N.A. | N.A. | EUR$ 30.000/ QALY Int$ 18.900/QALY | EUR$ 120 Int$ 77 | EUR$ 2.300 Int$ 1.492 |
| **Navani et al [23]** | Direct medical costs | Facility fees | Manufacturers prices, local hospital costs, NHS tariffs | 2010–2011 | British Pounds (GBP) and USD | N.A. | N.A. | N.A. | GBP$ 1.892 Int $ 1.492 | GBP$ 3.228 Int$ 2.535 |
| **Sharples et al [21]** | Direct medical costs | Staff time, bed occupancy, hospital fees, equipment costs (5-year lifetime), consumables, sterilization of scopes, maintenance contract | Standard treatment and procedures—NHS tariffs EBUS-TBNA and EUS-FNA —estimated by the Papworth Hospital finance department | 2008–2009 | British Pounds (GBP) | N.A. | N.A. | GBP$ 30.000 / QALY Int$ 20.670/QALY | GBP$ 10.808 Int$ 8.796 | GBP$ 11.735 Int$ 9.540 |
| **Søgaard et al [24]** | Direct medical costs | Costs of procedures, costs of treatment (surgical and nonsurgical regimen) | National average tariffs of the DRG system | 2010 | Euro (EUR) | adjusted to 2010 | 3% | N.A. | EUR$ 19.933 Int$ 19.590 | EUR$ 20.803 Int$ 20.445 |

(*Continued*)

**Table 2.** (Continued)

| Author | Type of Costs | Cost Items | Cost Data Sources | Year Accounted | Currency Unit | Inflation Rate | Discount Rate | *Willingness-to-pay Threshold | *EBUS-TBNA $ | *Mediastinoscopy $ |
|---|---|---|---|---|---|---|---|---|---|---|
| **Steinfort et al [27]** | Direct medical costs | Procedures costs | Primary data (actual patient data at the Royal Melbourne Hospital) | 2007–2008 | Australian Dollar (AUD) | 3% | N.A. | N.A. | AUD$ 1.318 Int$ 2.290 | AUD$ 5.324 Int$ 9.212 |

DRG, diagnosis-related groups; NSCLC, non-small cell lung cancer; EBUS-TBNA, endobronchial ultrasound-guided transbronchial needle aspiration; EUS-FNA, endoscopic ultrasound-guided fine-needle aspiration; QALY, quality-adjusted life years; N.A., not available

*currency unit conversion data: https://data.worldbank.org

cost savings for EBUS-TBNA mediastinal lung cancer staging strategy when compared to mediastinoscopy. Czarnecka-Kujawa and the group of the Toronto General Hospital calculated and used the ICER as outcome using a willingness-to-pay threshold of CAD$80.000/QALY [26]. The invasive staging strategy with EBUS-TBNA followed by mediastinoscopy offered the highest QALYs. In the cost comparison, the least expensive strategy was the "no invasive staging" strategy (patients sent directly to surgery without EBUS-TBNA or mediastinoscopy), followed by EBUS-TBNA, mediastinoscopy, EBUS-TBNA with confirmatory mediastinoscopy, EBUS-TBNA in the operating room and EBUS-TBNA in the operating room with confirmatory mediastinoscopy. The ICER was CAD$26.000 / QALY for EBUS-TBNA staging and CAD$1.400.000 / QALY for EBUS-TBNA followed by mediastinoscopy in case of negative findings after EBUS-TBNA. The mediastinoscopy strategy was dominated. Data from the ASTER Trial published by Sharples and colleagues showed no significant differences in

**Table 3. Interventions, comparators and outcomes.**

| Author | Intervention | Comparators | Outcomes |
|---|---|---|---|
| **Ang et al [28]** | EBUS-TBNA | Mediastinoscopy | Cost-savings per positive mediastinal lung cancer staging |
| **Czarnecka-Kujawa et al [26]** | EBUS-TBNA | No invasive staging / Mediastinoscopy / EBUS-TBNA followed by mediastinoscopy if negative result / EBUS-TBNA in OR | ICER for mediastinal staging of clinical N0/N1 lung cancer |
| **Harewood et al [25]** | EBUS-TBNA | Mediastinoscopy / TBNA / EUS-FNA / EBUS + EUS-FNA / combined EUS-FNA and TBNA / combined EBUS-TBNA and TBNA | Cost-savings for mediastinal lung cancer staging |
| **Luque et al [22]** | EBUS-TBNA | Thorax CT / PET-CT / Mediastinoscopy / TBNA / EUS-FNA | Optimal sequence of tests for mediastinal staging of NSCLC |
| **Navani et al [23]** | EBUS-TBNA | Mediastinoscopy | Cost-savings for isolated mediastinal lymphadenopathy diagnostic |
| **Sharples et al [21]** | EBUS-TBNA and EUS-FNA followed by mediastinoscopy if negative result | Mediastinoscopy | Cost-utility for mediastinal lung cancer staging |
| **Søgaard et al [24]** | EBUS-TBNA | Mediastinoscopy / EUS-FNA / PET-CT >> N2 or N3 >> EBUS-TBNA / PET-CT >> EBUS-TBNA | Cost for life-year gained for mediastinal lung cancer staging |
| **Steinfort et al [27]** | EBUS-TBNA | Mediastinoscopy / EBUS-TBNA followed by mediastinoscopy if negative results / TBNA followed by mediastinoscopy if negative results | Cost-savings for mediastinal lung cancer staging |

EBUS-TBNA, endobronchial ultrasound-guided transbronchial needle aspiration; EUS-FNA, endoscopic ultrasound-guided fine-needle aspiration; NSCLC, non-small cell lung cancer; OR, operating room; TBNA, transbronchial needle aspiration; PET-CT, positron emission computed tomography; ICER, incremental cost-effectiveness ratio

**Table 4. Cost-effectiveness, cost-savings and sensitivity analysis results.**

| Author | Cost-effectiveness and Cost-savings results | Sensitivity analysis results | Conclusions |
|---|---|---|---|
| **Ang et al [28]** | EBUS-TBNA resulted in SGD$ 1.214 cost savings per positive staging of lung cancer as compared to mediastinoscopy | EBUS is less costly than mediastinoscopy provided the sensitivity of EBUS is > 74% | EBUS-TBNA could result in cost savings per positive lung cancer staging compared to mediastinoscopy |
| **Czarnecka-Kujawa et al [26]** | The ICER of EBUS-TBNA compared to OR (no invasive staging) is 26.000/QALY | One-way: EBUS-TBNA is cost-effective between MLNM prevalence of 2.5% and 57% / EBUS-TBNA is cost-effective if its sensitivity is > 25% Two-way: Mediastinoscopy becomes cost-effective if the MLNM >11% and EBUS-TBNA sensitivity < 20% / Mediastinoscopy should be added after a negative EBUS if the MLNM is around 25% and sensitivity of EBUS around 60% | EBUS-TBNA staging in patients with N0 or N1 clinical nodal staging is cost-effective / Performing EBUS-TBNA in the operating room is not cost-effective |
| **Harewood et al [25]** | Initial EUS-FNA is the most economical strategy (USD$ 18.603) compared to EBUS-TBNA (USD$ 19.828) and mediastinoscopy (USD$ 20.157) | One-way: EUS-FNA remained the least costly strategy provided MLNM prevalence < 32%, above this prevalence, combined EUS and EBUS-TBNA is the most economical approach / EUS-FNA is least costly if its sensitivity remains > 50%, EBUS-TBNA becomes least costly if its sensitivity > 71% Two-way: throughout all FNA sensitivities EUS-FNA is the preferred strategy with MLNM prevalence > 32%, above this, the combination of EUS and EBUS-TBNA is the approach of choice | EUS-FNA is the least expensive strategy for mediastinal lung cancer staging when N2 probability <32% / EUS + EBUS-TBNA is least expensive when N2 probability> 32% |
| **Luque et al [22]** | Considering a willingness to pay of EUR $30.000/QALY: a positive CT should be followed by a TBNA EBUS-TBNA should be done if the CT or the TBNA is negative | The resulting strategy is robust to the uncertainty of the numerical parameters | Positive chest CT findings should be followed by TBNA / Negative chest CT findings should be followed by EBUS-TBNA |
| **Navani et al [23]** | The mean cost savings per patient undergoing EBUS-TBNA compared to mediastinoscopy is GBP$ 1336 | N.A. | EBUS-TBNA presents cost savings when used as an initial strategy to evaluate isolated mediastinal lymphadenopathy |
| **Sharples et al [21]** | There was no significant difference in expected costs between the two strategies. The mean difference in QALYs was 0.015 in favor of the endosonography arm (with surgical staging if negative) | Scenario without confirmatory mediastinoscopy after a negative endosonographic result: the distribution of cost-effectiveness is shifted in favor of endosonography, so that the probability that endosonography alone is cost-effective is approximately 90% | EBUS-TBNA and EUS-FNA followed by mediastinoscopy strategy was more sensitive, with lower negative predictive value and avoided unnecessary thoracotomies, showing a slight improvement in effectiveness (without statistical significance) |
| **Søgaard et al [24]** | PET-CT followed by EBUS-TBNA for positive findings was the least expensive strategy Thorax-CT followed by EBUS-TBNA strategy showed a better relationship of life-years gained | Alternative scenario analysis (5% lower prevalence of distance metastases / 5% poorer test performance of PET-CT / all survival quality-adjusted by a factor of 0.70 / 20% higher costs of PET-CT) confirmed the high probability of the strategy of PET-CT followed by EBUS-TBNA for positive findings to be the optimal choice | The recommendation for the National Health Service policy-making in Denmark is to make combined PET-CT and EBUS-TBNA available for the staging of patients with NSCLC |
| **Steinfort et al [27]** | Initial evaluation with EBUS-TBNA (negative results surgically confirmed) was found to be the most cost-beneficial approach (AUD$ 2961) in comparison to EBUS-TBNA not surgically confirmed (AUD$ 3344), conventional TBNA (AUD$ 3754) and mediastinoscopy (AUD$ 8859) | One-way: EBUS-TBNA remained the least costly approach down to an MLNM prevalence of 30% / EBUS-TBNA not surgically confirmed is least costly provided EBUS sensitivity >93% Two-way: EBUS-TBNA remained the least costly approach across plausible ranges of MLNM prevalence and EBUS sensitivity | EBUS-TBNA with surgical confirmation of negative results is the least expensive modality for mediastinal lung cancer staging |

EBUS-TBNA, endobronchial ultrasound-guided transbronchial needle aspiration; EUS-FNA, endoscopic ultrasound-guided fine-needle aspiration; NSCLC, non-small cell lung cancer; OR, operating room; TBNA, transbronchial needle aspiration; PET-CT, positron emission computed tomography; ICER, incremental cost-effectiveness ratio; MLNM, mediastinal lymph node metastasis; N.A., not available

expected costs between the endosonographic and surgical strategies [21]. The authors estimate that for a willingness-to-pay threshold of GBP$30.000/QALY, there was a 91% chance that endosonography strategy compared with surgical staging strategy would be cost-effective.

According to the Danish study [24], two strategies for mediastinal lymph node staging of lung cancer dominated the others: [1] referring all patients to PET-CT, with confirmation of positive findings on central or contralateral nodal involvement by EBUS-TBNA and [2] sending all patients directly to EBUS-TBNA. The ICER associated with moving from PET-CT followed by EBUS-TBNA strategy to the EBUS-TBNA as the initial strategy was estimated at EUR$188,461 per life year. The dominated strategies included sending all patients to mediastinoscopy and sending all patients to EUS-FNA, as these strategies provided poorer outcomes at higher costs. Luque et al reported that, for a willingness-to-pay threshold of EUR$30.000/QALY, optimally a positive CT scan should be followed by TBNA and the EBUS should be performed only when the CT scan or the TBNA is negative [22]. According to this study, PET is never cost-effective for this willingness-to-pay threshold.

### Quality assessment

The risk of bias assessment was based on the CHEERS tool [18]. Supporting information session of this article shows all 24 checkpoints contemplated by the instrument. The eight articles included in this review were qualitatively evaluated as follows: symbolized as √ for each item fulfilled in full, as ≠ for each item partially fulfilled and as X for each item not fulfilled. For a better visual identification of the quality analysis in the presented table, the fulfilled items were marked green, the items partially fulfilled are in yellow and the ones not attended in red. If the checkpoint did not apply to the study in question, it was not considered in the quality assessment (symbolized as N.A.) and left blank. Fig 2 summarizes the proportion of articles that completely, partially or did not meet the different quality assessment items. Of note are the publications of Czarnecka-Kujawa [26], Sharples [21], Søogard [24] and Steinfort [27], with > 85% of the items fulfilled in full. Data regarding time-horizon, discount rate, funding source and potential conflict of interest were missing in most studies.

### Discussion

Systematic reviews of health economic evaluations are valuable to inform the development of new economic models, to study different strategies in other contexts, to identify the most relevant studies for a particular decision, and to identify the implicated economic trade-offs [29]. Research from the last years proposes methods to guide authors writing those specific kinds of systematic reviews [30–32]. Still, the generalizability of results stemming from different contexts represents a major challenge [29].

Some limitations were identified while conducting this systematic review. Initially, although the number of references found at the initial stage of the search process was high, few economic evaluation studies on the EBUS-TBNA technique were identified, and by restricting the inclusion of studies to full economic analysis, only eight articles were included at the end. It was possible to improve the cost comparison between the different studies after adjusting for inflation rates and conversion to the international dollar. However, as the composition of the cost items was quite heterogeneous among the articles (from simply the cost of the procedure to a more complete total cost composed of equipment, maintenance, labor, and complication treatment values), the comparison between studies and a synthesis of results were harder to achieve. Not all publications evaluated the use of the EBUS-TBNA technique for patients with suspected N2 / N3 disease. Czarnecka-Kujawa et al [26] evaluated patients with clinical N0 / N1 disease, and Navani et al [23] studied the technique for diagnosis of isolated mediastinal lymphadenopathy, two distinct clinical situations. The study perspective varied from a broad health system perspective to a local hospital perspective. We did not identify studies from Latin America or Africa that fulfilled the inclusion criteria for this systematic review,

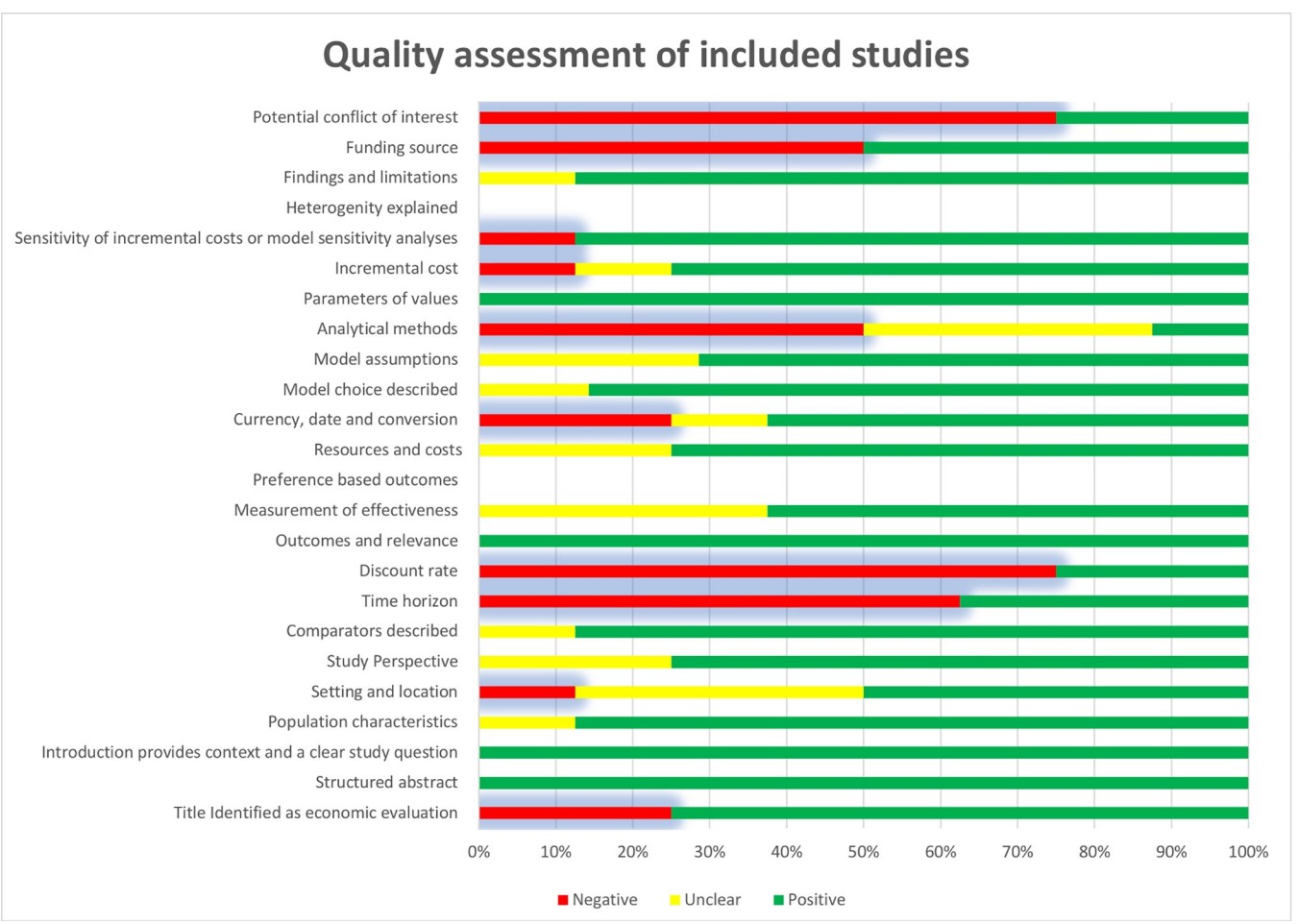

**Fig 2. Proportion of articles that filled the CHEERS quality assessment items.** *Items "Heterogeneity explained" and "Preference-based outcomes" were not available and left blank.

reinforcing the importance of conducting economic evaluation studies in these locations, especially because of unfavorable economic conditions and the differentiated prevalence of infectious diseases such as tuberculosis, which can alter mediastinal findings of patients with suspected lung cancer [33].

The risk of bias across studies is particularly relevant when a systematic review combines evidence on treatment effects across multiple studies. However, our review seeks to evaluate the results, methodological and reporting quality of economic evaluation studies, rather than the effect of any particular intervention, and did not combine results across studies. Tools for assessing publication or selective reporting bias (i.e. funnel plots) have been designed for examining the treatment effect of interventions, which cannot be applied to our study. We minimized publication bias by searching available protocols for economic evaluation studies of EBUS versus mediastinoscopy in systematic review registries available (ie. Prospero database). Additional assessment of the risk of bias across studies was also based on evaluations of each study's funding source and the nature of the disclosed conflict of interest for each study.

Despite these difficulties, this review presents relevant findings. The four cost-minimization analyses [23, 25, 27, 28] points to cost reductions related to the EBUS-TBNA strategy of mediastinal staging of lung cancer when compared to the surgical strategy. Two of the best quality

scored studies, the cost-utility publication of Czarnecka-Kujawa [26] and the cost-effectiveness analysis of Søogard [24], demonstrate that the mediastinoscopy strategy is dominated by the EBUS-TBNA strategy. Local EBUS sensitivity and the prevalence of MLNM can help to decide if EBUS should be the first staging strategy used and if a negative EBUS should be surgically confirmed or not. Additionally, the results suggest that the costs related to EBUS are higher when the procedure is performed in the operating room and this difference may have a negative impact on the cost-effectiveness of the test. According to Harewood et al 25), the EUS technique would be the most economical strategy for invasive mediastinal staging considering an MLNM prevalence of < 32%. Since the objective of this systematic review was to compare EBUS-TBNA and mediastinoscopy and not all studies evaluated the use of EUS, it is not possible to conclude from the collected data whether a strategy using EUS as the initial invasive staging examination would be more cost-effective. It is also important to note that the use of EUS has the limitation of not evaluating hilar lymph nodes, which may be important for defining the most appropriate therapeutic strategy in some cases. However, it is safe to conclude from the studies that a minimally invasive endosonographic staging strategy is associated with lower costs than surgical staging.

Finally, although not addressed in the studies evaluated by this systematic review, it is important to note that the EBUS technique is highly operator dependent and the results of a service with little experience can differ greatly from those published in the literature by experts. A complete mediastinal staging approach with systematic sampling of all multiple lymph node stations is quite different and more difficult to do than just sampling one suspected lymph node. Ensuring adequate training and quality control of the results obtained by EBUS is essential for the establishment of reference centers in the technique [34]. In this context, although evidence is insufficient to recommend that rapid onset evaluation (ROSE) should be used in every procedure [35], the presence of the pathologist in the bronchoscopy room can be of great value in guiding less experienced operators in obtaining representative lymph node samples. In cases where the clinical suspicion of mediastinal node involvement remains high after a negative result using a needle technique, surgical staging with mediastinoscopy should be considered [3].

## Conclusion

The information gathered in the eight different studies of this systematic review suggest that EBUS-TBNA is cost-effective compared to mediastinoscopy for mediastinal staging of lung cancer. The more comprehensive assessment of cost items related to the EBUS-TBNA strategy presented by the HTA published by Sharples et al may be useful as a starting point for future health economic evaluations of this procedure. Local EBUS-TBNA sensitivity, MLNM prevalence, and procedure site (inside or outside the operating room) are parameters with the greatest impact on the results of the cost-effectiveness of the method. Although this review brings important information from the current literature on the subject, economic studies considering their contexts in different countries should be conducted to guide decision-making by the respective health systems.

## Supporting information

**S1 Checklist. PRISMA Checklist.** PRISMA 2009 checklist.
(DOC)

**S1 File. Search Keys.** Search Keys used based on the PICO strategy.
(PDF)

**S2 File. Search strategy.** Search strategy used on Medline (Pubmed).
(PDF)

**S3 File. Quality assessment.** Publications evaluated according to the CHEERS checkpoint.
(PDF)

**S1 Data. Data abstraction form.** Data abstraction form used for data abstraction.
(DOCX)

## Author Contributions

**Conceptualization:** João Pedro Steinhauser Motta, Ricardo E. Steffen.

**Data curation:** João Pedro Steinhauser Motta, Caroliny Samary Lobato, Vanessa Souza Mendonça.

**Funding acquisition:** José Roberto Lapa e Silva.

**Investigation:** João Pedro Steinhauser Motta, Caroliny Samary Lobato, Vanessa Souza Mendonça.

**Methodology:** João Pedro Steinhauser Motta, Ricardo E. Steffen.

**Project administration:** José Roberto Lapa e Silva.

**Supervision:** Ricardo E. Steffen, José Roberto Lapa e Silva.

**Writing – original draft:** João Pedro Steinhauser Motta.

**Writing – review & editing:** João Pedro Steinhauser Motta, Ricardo E. Steffen, José Roberto Lapa e Silva.

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
