## [Decision Letter · Decision Letter 0]

13 Mar 2020

PONE-D-20-00791

ENDOBRONCHIAL ULTRASOUND-GUIDED TRANSBRONCHIAL NEEDLE ASPIRATION VERSUS MEDIASTINOSCOPY FOR MEDIASTINAL STAGING OF LUNG CANCER: SYSTEMATIC REVIEW OF ECONOMIC EVALUATION STUDIES

PLOS ONE

Dear Dr Steinhauser Motta,

Thank you for submitting your manuscript to PLOS ONE. In this study, Mota, et al. performed a systematic review of 8 studies evaluating the cost effectiveness of EBUS compared to mediastinoscopy in lung cancer. The study is of interest to readers. Several questions and comments were raised by the reviewers. Please consider these suggestions thoughtfully as they are meant to be constructive. After careful consideration, we feel that it has merit but does not fully meet PLOS ONE’s publication criteria as it currently stands. Therefore, we invite you to submit a revised version of the manuscript that addresses the points raised during the review process.

The authors should consider including a funnel plot to their results. 

The cost-effectiveness and cost-savings data should be combined in a separate table.

While the authors demonstrate that EBUS is cost-effective compared to mediastinoscopy, the potential limitations of EBUS should be included in the discussion. EBUS is operator dependent and full mediastinal staging with systematic sampling of multiple lymph node stations should be encouraged rather than FNA of a single lymph node. Training and quality control with rapid on-site evaluation (ROSE) are also important for those early in their experience with EBUS. There also continues to be a role for mediastinoscopy in cases where EBUS is non-diagnostic or there remains a high suspicion of positive mediastinal disease despite a negative EBUS.

Please have the paper reviewed by a native English speaker. There are several typographical and grammatical errors in the manuscript as pointed out by the reviewers.

We would appreciate receiving your revised manuscript by Apr 27 2020 11:59PM. To enhance the reproducibility of your results, we recommend that if applicable you deposit your laboratory protocols in protocols.io, where a protocol can be assigned its own identifier (DOI) such that it can be cited independently in the future. For instructions see: http://journals.plos.org/plosone/s/submission-guidelines#loc-laboratory-protocols

We look forward to receiving your revised manuscript.

Kind regards,

Jules Lin, M.D.

Academic Editor

PLOS ONE

Journal Requirements:

Reviewers' comments:

Reviewer's Responses to Questions

**Comments to the Author**

1. Is the manuscript technically sound, and do the data support the conclusions?

Reviewer #1: Yes

Reviewer #2: Yes

2. Has the statistical analysis been performed appropriately and rigorously? 

Reviewer #1: No

Reviewer #2: Yes

3. Have the authors made all data underlying the findings in their manuscript fully available?

Reviewer #1: Yes

Reviewer #2: Yes

4. Is the manuscript presented in an intelligible fashion and written in standard English?

Reviewer #1: No

Reviewer #2: Yes

5. Review Comments to the Author

Reviewer #1: I would like to thank the authors for allowing us to read their manuscript. This systematic review article evaluates and compares the cost /effectiveness of both EBUS vs mediastinoscopy in patients with lung cancer. The authors have selected an important and difficult area for study. The main goal of this paper is to evaluate the economic impact of EBUS and compared to mediastinoscopy. I do not find problems with this study ethics. Funding is clearly stated.

Manuscript Strengths:

- Abstract: Well presented, see minor comments

- Introduction: Overall, well presented. See comments

- Material and methods. I find it focused, with all inclusion/exclusion criteria and informative. There was a comprehensive literature search, the information sources were listed. Not redundant. See comments for suggestions. In regard to the data abstraction, it seems there was not structured data abstraction form used, however, there is description of the number of authors who abstracted the data, how they resolved disagreements, and some characteristics of all the studies used (See comments).

- Results: Needs to be re-organized for the reader to follow. See comments

- Discussion. The discussion is as well-balanced with mention made of some obvious limitations associated with the heterogeneity’s of the studies.

- Conclusions: Well written, clear.

I would like to raise the following major points:

- In the introduction section:

o Please consider including a clinically relevant and focused main study question. The objective is stated, but what is the authors main question to solve? This needs to be explicitly stated and will need to describe primary and secondary objectives.

o Please consider adding if the cost /effectiveness of EBUS vs mediastinoscopy has NOT been clearly demonstrated in prior clinical trials.

- In the material and methods:

o Please explain if a funnel plot was used

o Please consider adding

If the author had made attempts at collecting unpublished data

If a structured data abstraction form was used when selecting/scoring the studies

The size (n) of population for each study in Table 1.

- Results.

o There was repetition in the sub-section outcomes. results and sensitivity analysis’ paragraphs and Table 3. Please consider highlighting most relevant data. Tables and figures both need minor changes. See comments for suggestions.

o Could the authors consider cost-effectiveness and cost-savings description of data be be condensed on a separate table? And just include one or two shorter paragraphs to add some additional data/guide the reader?

- The discussion is very well written and succinct. It describes the major findings and acknowledges the limitations of this meta-analysis.

o There is an obvious trend of EBUS to be more cost effective when compared to mediastinoscopy. My major concern was that the cost items in all eight studies are very heterogenous. The authors elegantly describe how they reconciled this fact. I would like, however ask the authors if were all these eight studies, in their opinion, combinable? If so, please add this in the discussion. I noticed that a funnel plot was not used. Was this necessary, if not, was the sensitivity analysis enough? Please explain.

o Please consider mentioning that additionally, EUS will not be able to sample hilar lymph nodes but mediastinal).

Minor comments:

- Abstract. Correct misspelling “statement”. Consider changing keys for keywords, pulmonary neoplasia for lung cancer or thoracic malignancy.

- Line 153, misspelling “Computer” for “Computed”. Should read Chest computed tomography

- Line 180, please spell out DRG abbreviation

- Line 209, Please clarify if the authors meant EUS or EBUS or both.

- Line 269. Maybe a typo. I only found 8 articles used, but it is written “nine”

- Line 334. Delete “the”

- Table 1.

o Please correct the word “cost-effectiveness”, “available”, “months”

- Consider changing “thorax CT” for “Chest CT” throughout the document

- Table 3. Add in authors column the corresponding # of bibliography. Consider capital letter in sensitivity analysis column. Multiple misspellings: “n.a.” and “available”, computer for “computed”.

Summary:

In particular, this systematic review confirmed the cost effectiveness of EBUS when compared to mediastinoscopy on patients with thoracic malignancy. Clear limitations of the analysis are stated. I do think that the journal readers, will find this paper - once reviewed- helpful and thought-provoking for future health-economic research.

Reviewer #2: The authors conducted a systematic review of the economic evaluation of EBUS vs mediastinoscopy in a contemporary time period. They ultimately included 8 studies in the review and overall this shows that EBUS is a cost-effective procedure for lung cancer staging compared to mediastinoscopy. This is a relevant and interesting topic, as some places in the world do not have full access to this technology.

I have a few comments for the authors to help strengthen the manuscript.

1. Abstract intro: EBUS did not change the approach of pulmonary neoplasia. It has changed the approach to staging pulmonary neoplasia. This should be corrected throughout the manuscript.

2. There are several typos throughout the manuscript that need to be corrected prior to publication.

3. I assume the column heading in Table 1 should read "Sensitivity Analysis" and not Sensitive

4. The discussion would be strengthened by mention of the potential limitations of EBUS, in that it is highly operator dependent, ie the time spent doing the procedure by an experienced operator systematically sampling all nodal stations will likely have significantly different outcomes in terms of nodal sampling rate, etc than one done by an inexperienced operator only sampling one node. While this is not directly related to the findings from the included studies, it could impact cost ultimately. Given that most older experienced thoracic surgeons are more familiar with mediastinoscopy than EBUS, I think the inclusion of some discussion about the importance of adequate training and quality control of EBUS, including Rapid On-site Evaluation (ROSE) of specimens should be included, even if not mentioned in the included studies.

Overall, I think the manuscript is well put together and I congratulate the authors on their work.

6. PLOS authors have the option to publish the peer review history of their article (what does this mean?). If published, this will include your full peer review and any attached files.

Reviewer #1: No

Reviewer #2: No

---

## [Author Response · Author response to Decision Letter 0]

10 Apr 2020

Dear academic editor and reviewers, 

 Thank you very much for taking the time to review the article “Endobronchial ultrasound-guided transbronchial needle aspiration versus mediastinoscopy for mediastinal staging of lung cancer: systematic review of economic evaluation studies” submitted to Plos One. I am sure that the comments and suggestions made will improve the scientific quality of the article and interest for future readers.

1. Responses to the points raised by the academic editor:

• The authors should consider including a funnel plot to their results – the following paragraph was added to the manuscript on lines 311-321: “Risk of bias across studies is particularly relevant when a systematic review combines evidence on treatment effects across multiple studies. However, our review seeks to evaluate the results, methodological and reporting quality of economic evaluation studies, rather than the effect of any particular intervention, and did not combine results across studies. Tools for assessing publication or selective reporting bias (i.e. funnel plots) have been designed for examining treatment effect of interventions, which cannot be applied to our study. We minimized publication bias by searching available protocols for economic evaluation studies of EBUS versus mediastinoscopy in systematic review registries available (ie. Prospero database). Additional assessment of the risk of bias across studies was also based on evaluations of each study’s funding source and the nature of disclosed conflict of interest for each study.”

• The cost-effectiveness and cost-savings data should be combined in a separate table – Table 3 was divided into 2 tables (table 3 and table 4). Table 3 presents the interventions, comparators and outcomes of each study and Table 4 shows the results of cost-effectiveness, cost reduction and sensitivity analysis. Changes were made to the text of the manuscript with the inclusion of a sub-section with the title “Intervention, comparators and outcomes” and another with the title “Cost-effectiveness, cost-savings and sensitivity analysis results. The main information in the tables is described in these 2 subsections.

• While the authors demonstrate that EBUS is cost-effective compared to mediastinoscopy, the potential limitations of EBUS should be included in the discussion – the potential limitation of EBUS were included in the discussion as follows on line 342 to 354: “Finally, although not addressed in the studies evaluated by this systematic review, it is important to note that the EBUS technique is highly operator dependent and the results of a service with little experience can differ greatly from those published in the literature by experts. A complete mediastinal staging approach with systematic sampling of all multiple lymph node stations is quite different and more difficult to do than just sampling one suspected lymph node. Ensuring adequate training and quality control of the results obtained by EBUS is essential for the establishment of reference centers in the technique (34). In this context, although evidence is insufficient to recommend that rapid onset evaluation (ROSE) should be used in every procedure (35), the presence of the pathologist in the bronchoscopy room can be of great value in guiding less experienced operators in obtaining representative lymph node samples. In cases where the clinical suspicion of mediastinal node involvement remains high after a negative result using a needle technique, surgical staging with mediastinoscopy should be considered (3).”

• Please have the paper reviewed by a native English speaker. There are several typographical and grammatical errors in the manuscript as pointed out by the reviewers – the manuscript was reviewed by a native English speaker and the typographical and grammatical errors were corrected.

2. Responses to the points raised by Reviewer 1:

• In regard to the data abstraction, it seems there was not structured data abstraction form used - A structured data abstraction form was used and the form can now be accessed in the supporting information session. This information is now written in the manuscript as follows on lines 113-14: “A structured data abstraction form was used and can also be accessed in the supporting information session of this article.”

• Please consider including a clinically relevant and focused main study question. The objective is stated, but what is the authors main question to solve? This needs to be explicitly stated and will need to describe primary and secondary objectives – The main study question and description of primary and secondary objectives were included as follows on lines 69 – 71 “An important question to be answered at this point is: is the use of EBUS for the mediastinal staging of lung cancer cost-effective when compared to mediastinoscopy?”and 73 – 80 “The primary objective of this study is to understand the cost-effectiveness ratio of EBUS compared to mediastinoscopy for invasive mediastinal staging of lung cancer. Secondary objectives are to identify the most relevant economic evaluation studies published comparing EBUS to mediastinoscopy and what types of models were used in those evaluations, to understand the most important economic trade-offs and to guide future economic assessments to be carried out in countries with different health systems.”

• Please consider adding if the cost /effectiveness of EBUS vs mediastinoscopy has NOT been clearly demonstrated in prior clinical trials – This information was added to the manuscript on lines 73-74: “but until now the cost-effectiveness of EBUS versus mediastinoscopy has not been clearly demonstrated in prior clinical trials.”

• Please explain if a funnel plot was used - the following paragraph was added to the manuscript on lines 311-321: “Risk of bias across studies is particularly relevant when a systematic review combines evidence on treatment effects across multiple studies. However, our review seeks to evaluate the results, methodological and reporting quality of economic evaluation studies, rather than the effect of any particular intervention, and did not combine results across studies. Tools for assessing publication or selective reporting bias (i.e. funnel plots) have been designed for examining treatment effect of interventions, which cannot be applied to our study. We minimized publication bias by searching available protocols for economic evaluation studies of EBUS versus mediastinoscopy in systematic review registries available (ie. Prospero database). Additional assessment of the risk of bias across studies was also based on evaluations of each study’s funding source and the nature of disclosed conflict of interest for each study.”

• Please consider adding if the author had made attempts at collecting unpublished data – This information was added to the manuscript on lines 99-100 as follows: “The authors chose not to include unpublished data and gray literature in the searches.”

• Please consider adding if a structured data abstraction form was used when selecting/scoring the studies - A structured data abstraction form was used and the form can now be accessed in the supporting information session. This information is now written in the manuscript as follows on lines 113-14: “A structured data abstraction form was used and can also be accessed in the supporting information session of this article.”

• Please consider adding the size (n) of population for each study in Table 1 – The size (N) of population for each study was added in Table 1.

• There was repetition in the sub-section outcomes. results and sensitivity analysis’ paragraphs and Table 3. Please consider highlighting most relevant data – The authors agreed that there was an excess of repeated information between the sub-section outcomes, results and sensitivity analysis of the manuscript and table 3. Table 3 was divided in 2 Tables (Table 3 and Table 4) and the sub-section “Outcomes, results and sensitivity analysis” was divided in 2 sub-sections: “Intervention, comparators and outcomes” and “Cost-effectiveness, cost-savings and sensitivity analysis results”. The more detailed information was taken from the text and only the messages that we consider most important from each article were left to be highlighted. 

• Could the authors consider cost-effectiveness and cost-savings description of data be condensed on a separate table? And just include one or two shorter paragraphs to add some additional data/guide the reader? – Table 3 was divided into 2 tables (table 3 and table 4). Table 3 presents the interventions, comparators and outcomes of each study and Table 4 shows the results of cost-effectiveness, cost-savings and sensitivity analysis. Changes were made to the text of the manuscript with the inclusion of a sub-section with the title “Intervention, comparators and outcomes” and another with the title “Cost-effectiveness, cost-savings and sensitivity analysis results”. The main information in the tables is described in these 2 subsections.

• I would like, however ask the authors if were all these eight studies, in their opinion, combinable? If so, please add this in the discussion. I noticed that a funnel plot was not used. Was this necessary, if not, was the sensitivity analysis enough? Please explain - the following paragraph was added to the manuscript on lines 311-321: “Risk of bias across studies is particularly relevant when a systematic review combines evidence on treatment effects across multiple studies. However, our review seeks to evaluate the results, methodological and reporting quality of economic evaluation studies, rather than the effect of any particular intervention, and did not combine results across studies. Tools for assessing publication or selective reporting bias (i.e. funnel plots) have been designed for examining treatment effect of interventions, which cannot be applied to our study. We minimized publication bias by searching available protocols for economic evaluation studies of EBUS versus mediastinoscopy in systematic review registries available (ie. Prospero database). Additional assessment of the risk of bias across studies was also based on evaluations of each study’s funding source and the nature of disclosed conflict of interest for each study.”

• Please consider mentioning that additionally, EUS will not be able to sample hilar lymph nodes but mediastinal – This information was added to the manuscript on lines 337 – 340 as follows: “It is also important to note that the use of EUS has the limitation of not evaluating hilar lymph nodes, which may be important for defining the most appropriate therapeutic strategy in some cases.”

• Minor comments:

• Abstract. Correct misspelling “statement”. Consider changing keys for keywords, pulmonary neoplasia for lung cancer or thoracic malignancy – Correction and changes were made on lines 26 – 27 “The emergence of endobronchial ultrasound (EBUS) changed the approach to staging lung cancer” and 31 – 32 “…according to the PRISMA statement and registred on PROSPERO (CRD42019107901). Search keywords were elaborated considering descriptors…” of the manuscript.

• Line 153, misspelling “Computer” for “Computed”. Should read Chest computed tomography – Correction was made as follows now on lines 168-169: “…status based on chest computed tomography (CT) and positron emission tomography-computed tomography (PET-CT)…”

• Line 180, please spell out DRG abbreviation – The abbreviation is now spelled as follows on line 194: “to national tariffs and/or Diagnosis Related Groups (DRG) fees (21, 23-25).”

• Line 209, Please clarify if the authors meant EUS or EBUS or both – This sentence was removed from the text of the manuscript to leave only the most relevant information in the tables. Table 4 shows that: "EUS-FNA is the least expensive strategy for mediastinal lung cancer staging when N2 probability <32% / EUS + EBUS-TBNA is least expensive when N2 probability> 32%".

• Line 269. Maybe a typo. I only found 8 articles used, but it is written “nine” – This is corrected on line 266 as follows: “The eight articles included in this review were…”

• Line 334. Delete “the” – This is corrected on line 357 as follows: “…that EBUS-TBNA is cost effective compared to mediastinoscopy…”

• Table 1 - Please correct the word “cost-effectiveness”, “available”, “months” – These corrections were made in Table 1.

• Consider changing “thorax CT” for “Chest CT” throughout the document – This change was made throughout the document.

• Table 3 Add in authors column the corresponding # of bibliography. Consider capital letter in sensitivity analysis column. Multiple misspellings: “n.a.” and “available”, computer for “computed” – These corrections and changes were made in Table 3.

3. Responses to the points raised by Reviewer 2:

• Abstract intro: EBUS did not change the approach of pulmonary neoplasia. It has changed the approach to staging pulmonary neoplasia. This should be corrected throughout the manuscript – This correction was made throughout the manuscript as follows on line 26 “The emergence of endobronchial ultrasound (EBUS) changed the approach to staging lung cancer.” and 59 “...changed the approach to staging lung cancer...”

• There are several typos throughout the manuscript that need to be corrected prior to publication – the manuscript was reviewed by a native English speaker and the typographical and grammatical errors were corrected.

• I assume the column heading in Table 1 should read "Sensitivity Analysis" and not Sensitive – The column heading in Table 1 was corrected to “Sensitivity Analysis”.

• The discussion would be strengthened by mention of the potential limitations of EBUS, in that it is highly operator dependent, ie the time spent doing the procedure by an experienced operator systematically sampling all nodal stations will likely have significantly different outcomes in terms of nodal sampling rate, etc than one done by an inexperienced operator only sampling one node. While this is not directly related to the findings from the included studies, it could impact cost ultimately. Given that most older experienced thoracic surgeons are more familiar with mediastinoscopy than EBUS, I think the inclusion of some discussion about the importance of adequate training and quality control of EBUS, including Rapid On-site Evaluation (ROSE) of specimens should be included, even if not mentioned in the included studies - the potential limitation of EBUS were included in the discussion as follows on line 342 to 354: “Finally, although not addressed in the studies evaluated by this systematic review, it is important to note that the EBUS technique is highly operator dependent and the results of a service with little experience can differ greatly from those published in the literature by experts. A complete mediastinal staging approach with systematic sampling of all multiple lymph node stations is quite different and more difficult to do than just sampling one suspected lymph node. Ensuring adequate training and quality control of the results obtained by EBUS is essential for the establishment of reference centers in the technique (34). In this context, although evidence is insufficient to recommend that rapid onset evaluation (ROSE) should be used in every procedure (35), the presence of the pathologist in the bronchoscopy room can be of great value in guiding less experienced operators in obtaining representative lymph node samples. In cases where the clinical suspicion of mediastinal node involvement remains high after a negative result using a needle technique, surgical staging with mediastinoscopy should be considered (3).”

---

## [Decision Letter · Decision Letter 1]

17 Jun 2020

ENDOBRONCHIAL ULTRASOUND-GUIDED TRANSBRONCHIAL NEEDLE ASPIRATION VERSUS MEDIASTINOSCOPY FOR MEDIASTINAL STAGING OF LUNG CANCER: A SYSTEMATIC REVIEW OF ECONOMIC EVALUATION STUDIES

PONE-D-20-00791R1

Dear Dr. Steinhauser Motta,

We’re pleased to inform you that your manuscript has been judged scientifically suitable for publication and will be formally accepted for publication once it meets all outstanding technical requirements.

Kind regards,

Shawn Groth

Academic Editor

PLOS ONE

Additional Editor Comments (optional):

The authors have adequately responded to the critiques of the reviewers.

Reviewers' comments:

Reviewer's Responses to Questions

**Comments to the Author**

1. If the authors have adequately addressed your comments raised in a previous round of review and you feel that this manuscript is now acceptable for publication, you may indicate that here to bypass the “Comments to the Author” section, enter your conflict of interest statement in the “Confidential to Editor” section, and submit your "Accept" recommendation.

Reviewer #1: All comments have been addressed

Reviewer #2: All comments have been addressed

2. Is the manuscript technically sound, and do the data support the conclusions?

Reviewer #1: Yes

Reviewer #2: (No Response)

3. Has the statistical analysis been performed appropriately and rigorously? 

Reviewer #1: Yes

Reviewer #2: (No Response)

4. Have the authors made all data underlying the findings in their manuscript fully available?

Reviewer #1: Yes

Reviewer #2: (No Response)

5. Is the manuscript presented in an intelligible fashion and written in standard English?

Reviewer #1: Yes

Reviewer #2: (No Response)

6. Review Comments to the Author

Reviewer #1: I do thank the authors for providing thoughtful and clear answers to each of my suggestions. I am satisfied with your replies.

Reviewer #2: My comments and concerns from the review have been addressed. The manuscript is now stronger for these changes.

7. PLOS authors have the option to publish the peer review history of their article (what does this mean?). If published, this will include your full peer review and any attached files.

Reviewer #1: No

Reviewer #2: No

---

## [Editor Report · Acceptance letter]

19 Jun 2020

PONE-D-20-00791R1 

Endobronchial ultrasound-guided transbronchial needle aspiration versus mediastinoscopy for mediastinal staging of lung cancer: a systematic review of economic evaluation studies 

Dear Dr. Steinhauser Motta:

I'm pleased to inform you that your manuscript has been deemed suitable for publication in PLOS ONE. Congratulations! Your manuscript is now with our production department. 

Kind regards, 

on behalf of

Dr. Shawn Groth 

Academic Editor

PLOS ONE